# Factors Associated with Self-Reported Post/Long-COVID—A Real-World Data Study

**DOI:** 10.3390/ijerph192316124

**Published:** 2022-12-02

**Authors:** Anja Thronicke, Maximilian Hinse, Stefanie Weinert, Alexandra Jakubowski, Gerrit Grieb, Harald Matthes

**Affiliations:** 1Research Institute Havelhöhe, Kladower Damm 221, 14089 Berlin, Germany; 2Department of Intensive Care and Cardiology, Hospital Gemeinschaftskrankenhaus Havelhoehe, Kladower Damm 221, 14089 Berlin, Germany; 3Department of Plastic Surgery and Hand Surgery, Hospital Gemeinschaftskrankenhaus Havelhoehe, Kladower Damm 221, 14089 Berlin, Germany; 4Department of Gastroenterology, Hospital Gemeinschaftskrankenhaus Havelhoehe, Kladower Damm 221, 14089 Berlin, Germany; 5Charité—Universitätsmedizin Berlin, Corporate Member of Freie Universität Berlin and Humboldt-Universität zu Berlin, Medical Clinic for Gastroenterology, Infectiology and Rheumatology, Hindenburgdamm 30, 12200 Berlin, Germany

**Keywords:** COVID-19, long COVID, post/long COVID, PLC, post-acute COVID-19 syndrome, real-world data study, association factors

## Abstract

Evidence suggests that Post/Long-COVID (PLC) is associated with a reduced health-related quality of life, however little knowledge exists on the risk factors that contribute to PLC. The objective of this prospective real-world data study was to evaluate factors associated with PLC using national online survey data. Adjusted multivariable regression analyses were performed using the software R. Between 14 April and 15 June 2021, 99 registered individuals reported to have suffered from PLC symptoms and the most common PLC symptoms reported were fatigue, dyspnoea, decreased strength, hyposmia, and memory loss. The odds of individuals suffering from COVID-19-associated anxiety, hyposmia, or heart palpitations developing PLC were eight times (OR 8.28, 95% CI 1.43–47.85, *p* < 0.01), five times (OR 4.74, 95% CI 1.59–14.12, *p* < 0.005), or three times (OR 2.62, 95% CI 1.72–3.99, *p* < 0.01) higher, respectively, than of those who had not experienced these symptoms. Individuals who experienced fatigue while having COVID-19 were seven times more likely to develop PLC fatigue than those who had not (OR 6.52, 95% CI: 4.29–9.91, *p* < 0.0001). Our findings revealed that 13% of the individuals who had previously suffered from COVID-19 subsequently reported having PLC. Furthermore, COVID-19-associated anxiety, hyposmia, heart palpitations, and fatigue were, among others, significant determinants for the development of PLC symptoms. Hyposmia has not previously been reported as an independent predictive factor for PLC. We suggest closely monitoring patients with COVID-19-induced fatigue, heart palpitations, and anxiety, as these symptoms may be predictors of PLC symptoms, including fatigue.

## 1. Introduction

COVID-19 is a multi-organ disease that can—like other virulent coronavirus strains—persist and/or cause prolonged organ-specific sequelae beyond four weeks from the onset of symptoms. In accordance with the Cochrane Rehabilitation review, “one of the following four criteria can be used to diagnose Post/Long-COVID (PLC): (1) symptoms that persist from the acute COVID-19 phase or its treatment, (2) symptoms that have resulted in a new health limitation, (3) new symptoms that have occurred after the end of the acute phase but are understood to be a consequence of COVID-19 disease, (4) worsening of a pre-existing underlying condition” [1]. Evidence suggests that PLC is associated with a reduced health-related quality of life (HRQL) [2]. Up to 80% of patients may experience PLC [2]. The most prevalent PLC symptoms are chest pain, fatigue, dyspnoea, coughing, sleep disturbance, arthralgia, headache, and anosmia [2,3,4,5]. Gastrointestinal sequelae and gastric microbiome dysregulation, among others, may also be common symptoms of PLC [6]. Newer clinical research points towards neuro- and cardio-associated symptoms of PLC, including postural orthostatic tachycardia syndrome, indicating that the malfunction of the olfactory system, the nervus vagous, and the autonomic nervous system may play a significant role in PLC [7]. Even though most PLC symptoms lessen over time, they may last from several months to over a year [8], and some of them can even become chronic or manifest as an independent syndrome. As SARS-CoV2 is in the process of becoming endemic, it is anticipated that the socio-economic consequences for the healthcare system of every country in the world will be tremendous. For instance, in the U.S., PLC incurred health-care costs of $386 billion in wages, savings, and medical expenses in just one month [9]. Little is known about the diagnostic criteria, standard terminology, or risk factor of this condition, and the literature on HRQL in affected patients is sparse. In addition, the mechanisms of PLC have not yet been determined. The objective of the present study was to evaluate the risk factors associated with PLC using real-world online survey data.

## 2. Materials and Methods

### 2.1. Study Design

A prospective, longitudinal, real-world online survey was conducted of participants registered in Germany. Data collection took place between 14 April and 15 June 2021. Preliminary data collection for the calculation of the proportion of people with PLC took place between April 2021 and July 2022. The present study’s primary aim was to evaluate factors associated with PLC using data from the real-world online survey. The demographics and participants’ self-reported history of comorbidities, COVID-19, and PLC symptoms were retrieved from the COVID-19 real-world online survey database. The online survey was registered at the German Clinical Trials Register with the ID number DRKS00024800.

### 2.2. Participants and Enrolment

Participants were recruited through promotions on various platforms including the German media press, the Twitter newsfeed of the Institute for Social Medicine, Epidemiology and Health Economics at the hospital Charité—Universitätsmedizin Berlin, the Facebook site of the hospital Gemeinschaftskrankenhaus Havelhöhe Berlin, the study register website of the Charité—Universitätsmedizin Berlin, the distribution list of the Vivantes clinics of the Auguste Victoria clinics Berlin, the distribution list of the Goetheanum, and the Society of Anthroposophic Physicians. Furthermore, flyers with a quick response code and a website address for the survey were handed out at various COVID-19 vaccination centres in Berlin. Interested persons could then enter the study’s webpage and access further information about the study and take part.

### 2.3. Ethics

The online survey study was approved by the Charité—Universitätsmedizin Berlin’s ethics committee (EA1/05/21) and was conducted in accordance with the ethical standards of the 1964 Declaration of Helsinki. The study was also approved by the Charité—Universitätsmedizin Berlin’s institutional data protection board and quality management board. Written informed consent was obtained from all participants prior to study enrolment.

### 2.4. Survey 

Self-reported histories of comorbidities, COVID-19, and PLC symptoms were retrieved from the real-world COVID-19 online survey. The survey was developed by a multi-disciplinary team of physicians, researchers, and psychologists and consists of a structured, anonymous, self-administered questionnaire including 27 items capturing six themes: (a) socio-demographic characteristics, such as age, gender, BMI, (b) participant’s comorbidities, (c) COVID-19 symptoms, (d) PLC problems, (e) medication taken by the individuals against COVID-19 or PLC symptoms, and (f) HRQL.

### 2.5. Data Collection

Data was collected via the Research Electronic Data Capture (REDCap) data management platform version 10.6.14 hosted at the hospital Charité—Universitätsmedizin Berlin. Two separate installations were used to conduct the survey in compliance with data protection regulations. To follow-up and to prevent repeated submission, the consent to the study was stored together with the participants’ e-mail addresses separately from the survey data (pseudonymized storage). The participants were then assigned with their unique pseudonymized ID, so that duplicate registrations and participations could be filtered out.

### 2.6. Statistical Analyses

Continuous variables were described as median with interquartile range (IQR); categorical variables were summarised as percentages. *p*-values < 0.05 were considered significant. To address potential sources of bias, a multivariable regression analysis was performed using R software (Version 3.6.1, R Development Core Team, Vienna, Austria). The outcome was dichotomic PLC (yes/no), adjusting for age (in years), gender (male/female), BMI (in kg/m^2^), COVID-19 anxiety (yes/no), COVID-19 fatigue (yes/no), COVID-19 headache (yes/no), COVID-19 heart palpitations (yes/no), and COVID-19 memory loss (yes/no).

## 3. Results

In total, 3510 individuals were enrolled and completed all or part of the survey questions between 14 April and 15 June 2021. A total of 273 of the 3510 enrolled individuals (7.8%) reported having COVID-19 and 99 of those (36.3%) reported suffering from PLC symptoms. Of the 2472 COVID-19 cases self-reported between April 2021 and July 2022, 244 (13%) cases self-reported PLC symptoms.

On average three PLC symptoms (IQR: 2–3.5) were reported (Figure 1).

As shown above (Figure 1), 75% of participants reported suffering from one to three symptoms, 21% reported four to six, and 4% reported more than seven PLC symptoms. Table 1 shows the baseline characteristics of the individuals who reported having PLC symptoms. The median age was 51.5 years, 77.8% identified as female, more than half of the individuals had comorbidities (57.6%), and more than half had children (58.6%). On average, individuals with PLC weighed a normal amount as the median body mass index (BMI) was below 25 kg/m^2^ and higher than 18.5 kg/m^2^, Table 1.

The most common PLC symptoms were fatigue (*n* = 54, 54.5%), dyspnoea (*n* = 28, 28.3%), decreased strength (*n* = 16, 16.2%), decreased sense of smell (*n* = 15, 15.2%), and memory loss (*n* = 14, 14.1%) including punctual amnesia (*n* = 1, 1.0%), see Figure 2. Further highly prevalent symptoms included concentration problems (*n* = 10, 10.1%), heart problems (*n* = 13, 13.1%), a decreased sense of taste (*n* = 10, 10.1%), coughing (*n* = 9, 9.1%), muscle aches (*n* = 9, 9.1%), and headaches (*n* = 8, 8.1%) as well as lung problems (*n* = 4, 4.0%), see Figure 2. Symptoms such as anxiety including irritability (*n* = 7, 7.1%) and increased skin sensitivity, noise sensitivity, and sensitivity to temperature (*n* = 5, 5.1%) were also reported, as were depressive episodes or depression (*n* = 6, 6.1%) and insomnia (*n* = 4, 4.0%). Other interesting self-reported symptoms which occurred less frequently included word finding or lingual problems (*n* = 4, 4.0%), tinnitus (*n* = 3, 3.0%), pyrexia (*n* = 3, 3.0%), hair loss (*n* = 2, 2.0%), sensory disturbances (tingling) of the lip, hands, and feet (*n* = 2, 2.0%), the so-called ‘brain fog’ (*n* = 2, 2.0%), and intestinal inflammation (*n* = 1, 1.0%), see Figure 2. Adjusted multivariable logistic regression analysis revealed that individuals with the comorbidity asthma had higher odds (OR 1.62, 95% CI: 0.99–2.65, *p* = 0.05) of reporting COVID-19 symptoms in the registry. Male individuals had relevant lower odds of reporting COVID-19 symptoms in the registry (OR 0.65, 95% CI: 0.49–0.89, *p* = 0.006). Furthermore, an increased BMI or the comorbidity hypertension were not significantly associated with COVID-19 symptoms.

As a high number of clinical studies show a gendered dimension to PLC, we performed univariate logistic regression and found that individuals identifying as female were 1.7 times more likely to report PLC symptoms than individuals identifying as male which is statistically significant (OR 1.73, 95% CI: 1.06–2.83, *p* = 0.027). However, after performing a multivariate regression analysis adjusting for gender, age, BMI, and COVID-19 symptoms, our results showed that gender, as well as age and BMI, was not associated with higher or lower odds of developing any symptoms of PLC. However, individuals who identified as male did have slightly lower odds of developing strong PLC symptoms, although this was not deemed statistically significant. Individuals reporting COVID-19-associated anxiety, as shown by multivariate regression analysis, were eight times more likely to develop PLC than individuals without this symptom (OR 8.28, 95% CI 1.43–47.85, *p* < 0.01), see Figure 3. Those who experienced strong symptoms of hyposmia whilst having COVID-19 were five times more likely (OR 4.74, 95% CI 1.59–14.12, *p* < 0.005) to develop PLC symptoms than those who did not experience strong symptoms of hyposmia, see Figure 3. In addition, those with COVID-19-associated symptoms such as fatigue (OR 3.09, 95% CI 2.09–4.56, *p* < 0.001), heart palpitations (OR 2.62, 95% CI 1.72–3.99, *p* < 0.001), or heavy headache symptoms (OR 2.83, 95% CI: 1.03–7.78, *p* = 0.04) were three times more likely to develop PLC symptoms than those who did not report these symptoms, see Figure 3. Furthermore, the odds of individuals with COVID-19-associated memory loss developing any PLC symptoms were two times higher than those of individuals who had not experienced COVID-19-associated memory loss (OR 1.80, 95% CI 1.14–2.95, *p* = 0.012), see Figure 3. Interestingly, participants who experienced strong symptoms of loss of taste (anosmia) were less likely to develop any signs of PLC than those who had not (OR 0.20, 95% CI: 0.06–0.66, *p* = 0.008), see Figure 3.

Individuals who experienced COVID-19 fatigue were seven times more likely to develop PLC fatigue than those who had not (OR 6.52, 95% CI: 4.29–9.91, *p* < 0.0001), see Figure 4 and Figure 5. Also, individuals who reported having heart palpitations or anxiety whilst having COVID-19 were two times more likely to develop PLC fatigue than those who did not report these symptoms (heart palpitations: OR 1.79, 95% CI: 1.25–2.54, *p* = 0.002, anxiety: OR 1.56, 95% CI: 1.06–2.30, *p* = 0.02), see Figure 4 and Figure 5. Age, gender, or BMI were not associated with lower or higher possibilities of developing PLC fatigue.

COVID-19 fatigue was also associated with a five times higher risk of developing PLC dyspnoea (OR 5.24, 95% CI: 3.90–7.04, *p* < 0.0001), Figure 5 and Figure 6. Individuals with COVID-19 heart palpitation symptoms were two times more likely to experience PLC dyspnoea than those without (OR 1.79, 95% CI 1.28–2.53, *p* = 0.0007), Figure 5 and Figure 6. Age, gender, or BMI were not significantly associated with the occurrence of PLC dyspnoea. Individuals who identified as male were slightly less likely to experience PLC fatigue or PLC dyspnoea than individuals who identified as female, however this was not statistically significant, Figure 4 and Figure 6. COVID-19 fatigue was also deemed a strong predictor for the development of low resilience during PLC. The odds of individuals with COVID-19 fatigue developing low resilience during PLC were six times higher than those of individuals without it (OR 6.08, 95% CI: 3.63–10.19, *p* < 0.001), Figure 5.

We also performed adjusted multivariable regression analysis for further association factors of PLC symptoms. Adjusting for age and gender, COVID-19 muscle pain doubled the likelihood of an individual reporting PLC hyposmia (OR 2.06, 95% CI 1.13–3.76, *p* = 0.019) and COVID-19 hyposmia tripled the likelihood (OR 3.29, 95% CI 1.74–6.21, *p* = 0.0002). A statistically significant link between COVID-19 heart palpitations and PLC heart palpitations was determined (OR 2.70, 95% CI 1.42–5.11, *p* = 0.0024) and an even stronger link was found between COVID-19 muscle pain and PLC heart palpitations (OR 6.50, 95% CI 2.26–18.70, *p* = 0.0005), see Figure 5.

## 4. Discussion

There is little knowledge on the risk factors for PLC. Our findings revealed that self-reported symptoms of anxiety or hyposmia experienced by individuals during COVID-19 were independent risk factors for self-reported PLC. Furthermore, suffering from fatigue whilst having COVID-19 was shown to be a risk factor for subsequent PLC fatigue and PLC dyspnoea. We have further analysed certain risk factors associated with self-reported COVID-19.

Participants of our study were, on average, older and most identified as female. Our participants also reported more comorbidities, a lower BMI, and reported having less children than the average of the general population in Germany. The results of our study reveal that more than half of individuals with PLC reported fatigue, one third reported dyspnoea, and 15% reported decreased strength, a decreased sense of smell, or memory loss. This is in line with a meta-regression analysis, which was performed in accordance with the PRISMA guidelines, where the most common, persistent symptoms of PLC were reported to be fatigue, dyspnoea, and anosmia [4].

The proportion of individuals with self-reported COVID-19 who suffered from PLC symptoms calculated in our study (13%) was similar to that calculated in another observational study that found 10% of COVID-19 patients showed symptoms that persisted for 12 weeks or longer [8]. In line with our observation, another study on 558 patients who had recovered from COVID-19 found 13.3% had self-reported PLC symptoms that persisted for 28 days or more [10]. Among a number of other good quality publications on the incidence of PLC there is another observational study that was conducted during a time frame comparable to our study. It revealed that 22% of patients with laboratory confirmed COVID-19 had Long COVID and 9.9% had PLC [8]. According to a review from August 2021, the incidence of PLC ranged from 10–35% with even higher scores for hospitalized patients [11]. Thus, the percentage of individuals with PLC in our study is within the range of published observational studies.

On average, three PLC symptoms (interquartile range, IQR: 2–3.5) were reported in our study, a result which is slightly below the range of case series studies that have reported an average of 4 symptoms per patient (IQR: 2 to 5) in patients with COVID-19 before they were admitted to hospital [12]. Thus, it is anticipated that our study cohort may be healthier than that in the study on pre-hospitalised COVID-19 patients and therefore patients in our study may have reported less symptoms.

Our findings reveal that individuals reporting a neurological symptom associated with COVID-19 hyposmia were more likely to develop PLC. Hyposmia, in concert with anosmia, has been reported as a good predictor of COVID-19 infections with a risk ratio of 4.56 [13]. In our study, the odds of experiencing PLC were comparable, i.e., five times higher when developing COVID-19 related hyposmia. So far, hyposmia had not been reported as a predictor of PLC. In addition, the results of our study showed that individuals reporting COVID-19 related anxiety were more likely to develop PLC. It has been shown that the incidence of COVID-19 related psychiatric symptoms is high—at least 35% of COVID-19 patients had anxiety or depression—and that these symptoms can even persist after recovery from COVID-19 [13]. Interestingly, a higher cytokine interleukin (IL)1ß level was seen in COVID-19 patients with anxiety and/or depression [14], which indicates that neuroimmune alterations may have taken place as a result of the individual having the disease. In addition, a correlation was found between the severity of COVID-19 and/or the duration of hospitalization and depression. Anxiety symptoms may improve around 90 days after an individual recovers from COVID-19 according to some studies [15], but other studies suggest that depression and anxiety are also good predictors of longer lasting post-traumatic stress disorder (PTSD) [16]. It has been shown that up to 43% of COVID-19 patients suffer from PTSD [17] compared with 7–10% of the general population over the pandemic [18]. There are conflicting opinions surrounding the improvement of PTSD over time, and PTSD has been reported as the most prevalent long-term psychiatric morbidity after a SARS-CoV2 infection [19]. In addition, the risk of suicidal behaviour is high in individuals with post-COVID neurological, psychiatric, and physical illness [20]. Therefore, psychiatric morbidities such as anxiety and depression must be closely monitored during the COVID-19 recovery period.

Our results emphasised that self-reported COVID-19 anxiety is a predictor for self-reported PLC fatigue. A meta-analysis revealed that a very high number of studies (*n* = 32) reported that up to 87% of individuals suffered from PLC fatigue after recovering from acute COVID-19 [21]. However, the severity of fatigue improves over time [15,22]. Anxiety and other psychiatric comorbidities were already reported in literature as predictors of PLC fatigue, with those who identify as female more likely to report it [23,24,25]. Some authors did not find any correlation between the severity of acute COVID-19 and PLC fatigue [26] and some others found a correlation [22,27]. Several studies observed similarities between PLC fatigue and myalgic encephalomyelitis/chronic fatigue syndrome [28], the latter being linked to various known virus infections, such as the Epstein–Barr virus [29] or enterovirus [30].

Our study also revealed that individuals with self-reported COVID-19 fatigue symptoms were more likely to develop a PLC dyspnoea. A study retrospectively analysing the risk factors of 2915 PLC patients found that those who had a respiratory illness at the beginning of COVID-19 were 1.4 times more likely to experience PLC than those who had not [31]. However, an association between acute COVID-19 fatigue and PLC dyspnoea was not seen in previous studies.

The findings of our study reveal that fatigue during COVID-19 seems to be a predictor for fatigue, dyspnoea, and low resilience during PLC. This may be explained by the fact that it is one of the most prevalent symptoms reported by patients experiencing COVID-19. However, as shown by the multivariate regression analysis in our study, COVID-19 fatigue remains a significant determinant even when adjusted for age, gender, or other confounding factors. Other authors report a significantly higher prevalence of PLC fatigue in individuals who identify as female [32], however, our real-world data study did not confirm these findings. The link between general PLC symptoms reported by individuals who identified as female was significant in a univariate analysis but not after adjusting for age, BMI, and other COVID-19 symptoms.

Self-reported heart palpitation, alongside the COVID-19 symptom fatigue, seems to play a relevant role for PLC dyspnoea and fatigue. Postural tachycardia syndrome (POTS) could be the underlying factor for this self-reported symptom. POTS has increasingly been reported in association with COVID-19 and/or PLC and has been linked to the malfunctioning of the autonomic nervous system [7]. The treatment options for POTS as well as other PLC symptoms are limited [33] since only a low number of physicians have the expertise to properly treat it. This means that in the near future, significant healthcare resources may be required [34].

Limitations: The nature of our study does not allow us to indicate the exact time point or underlying diagnosis of the PLC symptomatology of the individuals. Furthermore, our study uses COVID-19 and PLC symptoms self-reported via an online survey-tool, which may be prone to over- or under-reporting as well as potential misdiagnoses. In addition, a self-selection bias has to be anticipated as it is common that individuals with symptoms are more likely to volunteer to fill out surveys concerning their condition. Also, since COVID-19 is known to have ethnic differences in clinical response it would have been beneficial to report on this variable. However, ethnicity was not included in the survey. Another limitation is that the survey did not include questions on SARS-CoV-2 variants. Nevertheless, we assume that the SARS-CoV-2 Alpha variant was the predominant one during the observational period of our study (as the predominant variants that were registered in Germany were the SARS-CoV-2 Alpha variant in April 2021 and the SARS-CoV-2 Delta variant by the end of June 2021). The results were collected at a time when vaccination against the SARS-CoV2 virus was carried out for the first time in Germany. It is therefore possible that a higher proportion of unvaccinated patients with severe COVID-19 and more severe PLC symptoms were registered during this period. However, during the preparation of the manuscript, we also gained access to additional preliminary data from a survey of our group conducted from April 2021 to July 2022 that recorded a higher proportion of individuals vaccinated against the virus. Thus, we were able to calculate the proportion of PLC individuals during this period (13%) which demonstrated a more realistic proportion of individuals with self-reported PLC symptoms.

The potential risk factors of PLC that are already known are older age, identifying as female, severe clinical status, a high number of comorbidities, hospital admission, and oxygen supplementation during the acute COVID-19 phase [4]. The findings of our study may add COVID-19 related hyposmia, anxiety, fatigue, heart palpitations, and memory loss as possible predicting factors for long-term sequelae and we suggest that COVID-19 patients with these symptoms are closely monitored.

## 5. Conclusions

Our findings reveal that 13% of individuals who have previously suffered from COVID-19 reported suffering from PLC. Self-reported COVID-19-associated anxiety, hyposmia, fatigue, heart palpitations, and memory loss were significant determinants for the development of self-reported PLC symptoms. Previously, hyposmia had not been reported as an independent predictive factor for PLC. We suggest that the close monitoring of patients with these COVID-19-induced symptoms may help with predicting long-term sequelae.

## Figures and Tables

**Figure 1 ijerph-19-16124-f001:**
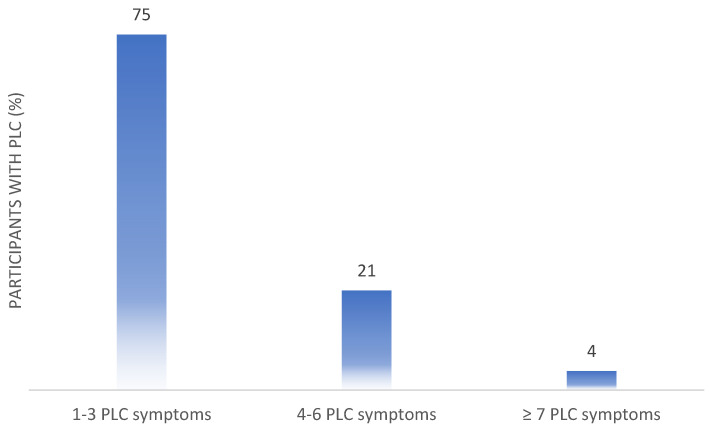
Proportion of participants with 1–3, 4–6 or ≥7 PLC symptoms; PLC, post-acute COVID-19 syndrome.

**Figure 2 ijerph-19-16124-f002:**
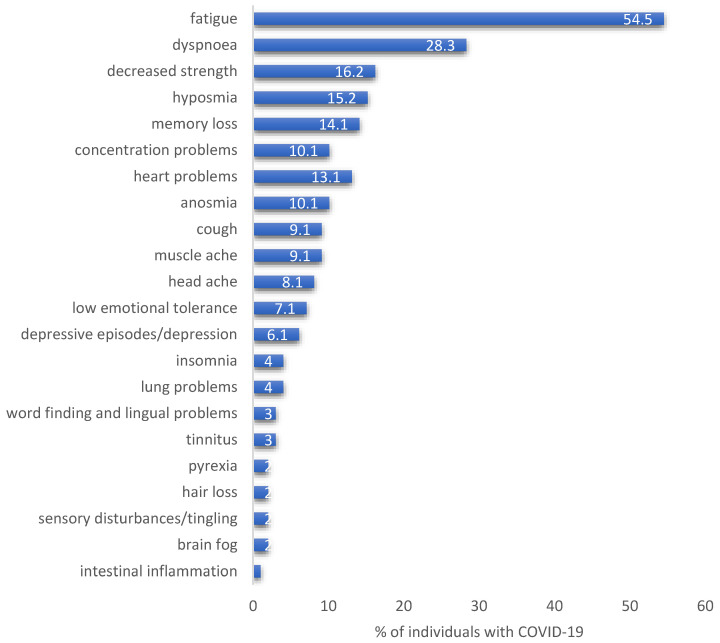
Self-reported PLC symptoms, proportion of individuals who have recently suffered from COVID-19, % percent.

**Figure 3 ijerph-19-16124-f003:**
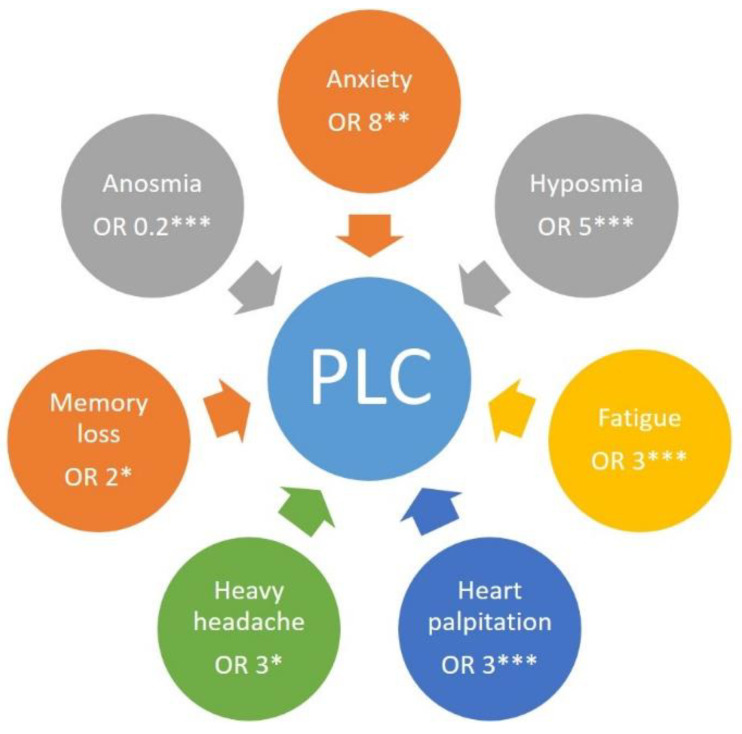
Self-reported Covid-19 related symptoms (outer coloured circles) with a relevant and significant probability for the development of PLC (inner blue circle) according to the results from adjusted multivariable logistic regression analysis. The number in the outer circles represents the probability (OR) of developing PLC, the stars represent the statistical significance. OR, odds ratio. * *p* < 0.05; ** *p* < 0.01; *** *p* < 0.005.

**Figure 4 ijerph-19-16124-f004:**
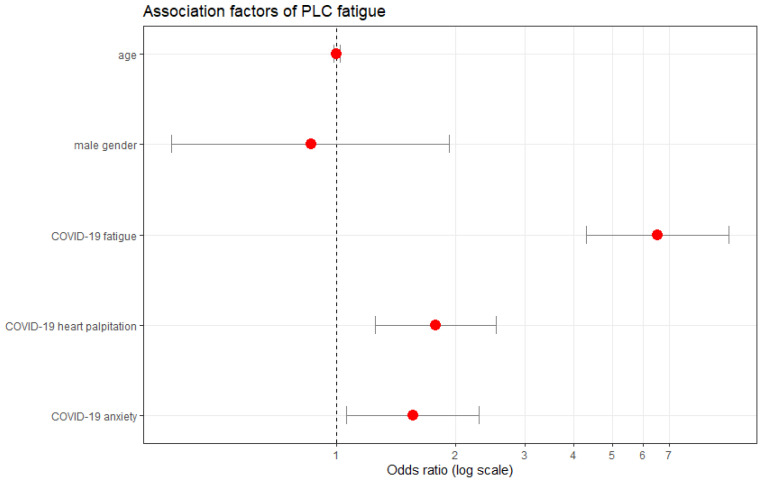
Probability of developing PLC fatigue, adjusted multivariable logistic regression analysis. Factors presented (except age and gender) are associated with a statistically significant (*p* < 0.05) reduced probability (left-hand side from the indicated margin) or increased probability (right-hand side from the indicated margin) of developing PLC fatigue; PLC, post-acute COVID-19 syndrome; red dots represent the probability (odds ratio) of developing PLC fatigue.

**Figure 5 ijerph-19-16124-f005:**
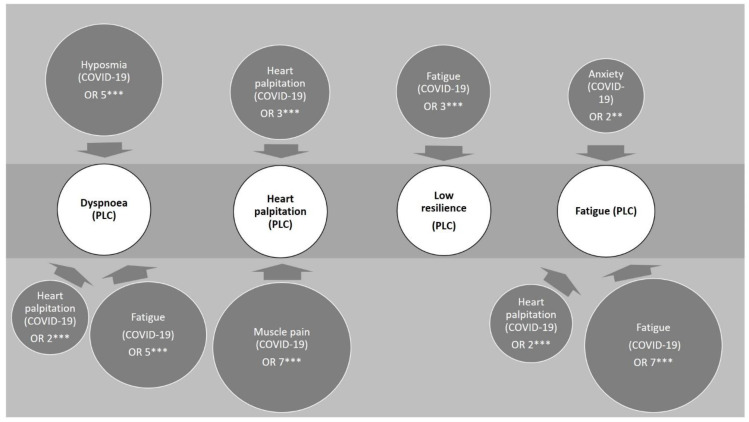
Flow chart for the association between self-reported COVID-19 symptoms (dark grey circles) and PLC symptoms (white circles). The number in the dark grey circles represents the probability (OR) of developing the relevant PLC symptom, the stars represent the statistical significance; OR, odds ratio, calculated from adjusted multivariable logistic regression analysis. ** *p* < 0.01; *** *p* < 0.005.

**Figure 6 ijerph-19-16124-f006:**
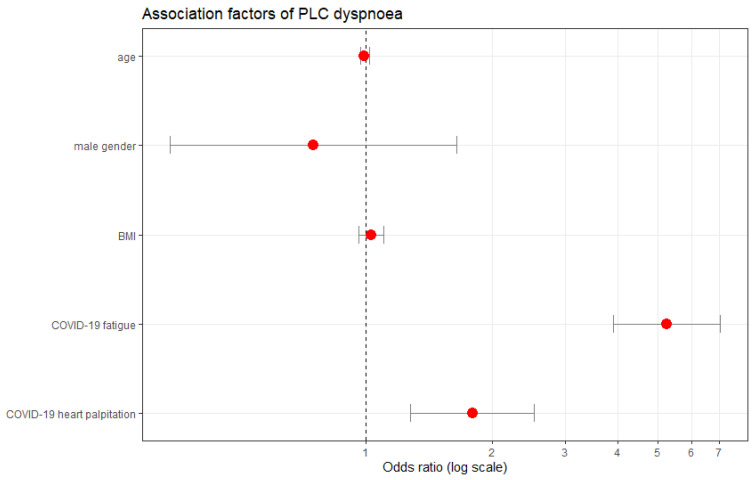
Probability of developing PLC dyspnoea, adjusted multivariable logistic regression analysis. Factors presented (except age, gender and BMI) are associated with a statistically significant (*p* < 0.05) reduced probability (left-hand side from the indicated margin) or increased probability (right-hand side from the indicated margin) of developing PLC dyspnoea; PLC, post-acute COVID-19 syndrome; BMI, body mass index; red dots represent the probability (odds ratio) of developing PLC dyspnoea.

**Table 1 ijerph-19-16124-t001:** Baseline characteristics of participants reporting PLC symptoms, *n* = 99.

	N	%
Age, median (IQR) in years	51.5	(39–60)
Comorbidities	57	57.6%
Gender, female	77	77.8%
Body Mass Index (BMI), median (IQR) in kg/m^2^	23.9	(22.1–27.8)
Has a child/children, yes	58	58.6%

## Data Availability

The anonymized data that support the findings of this study are openly available in the repository figshare.com, DOI: 10.6084/m9.figshare.21370086, https://figshare.com/articles/dataset/pcs/21370086 (accessed on 1 December 2022).

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
