# Peer review of "Factors Associated with Self-Reported Post/Long-COVID—A Real-World Data Study"

_ijerph, 2022, doi:10.3390/ijerph192316124_

Round 1

Reviewer 1 Report

Overall, the manuscript needs to be editing to limit the usage of phrases as "signficant X-times higher". Also, a lot of statistical data needs to be rather well represented in figures than in text format. It is hard to keep track of especially in a statistical study like this. 
While age and gender were considered to be factors for PLC, ethnicity wasnt reported. COVID-19 is known to have ethnic differences in clinical response and it would be good to see that data added. Similarly, it would be good to see differences in clinical response based on the specific COVID19 variant, if such data is available. 
Please proofread before resubmission of manuscript. 

Reviewer 2 Report

The text has several grammatical and spelling errors and unclear, poorly structured sentences.

A few examples:

Abstract: 

line 20: real-world instead of real.world

lines 25,26 and 28: 8 times (or 8-times); 3 times (or 3-times); 7 times (or 7-times)

line 33: "to closely monitor..." instead of "closely monitoring..."

Introduction:

line 50: "may be as well common" instead of "may be as well be common"

line 51-53 : sentence is difficult to understand and is grammatically incorrect with spelling errors

Results:

line 119-124: again, grammatically incorrect and incoherent sentences

line 156, 162,165, 167, 171, 184, 251: X times or X-times

line 187: "were associated" instead of "was associated"

Discussion:

line 230: the authors probably meant to write "...the average of the German population"

line 238: how about other observational studies? Did they report similar results?

lines 236-240: these sentences could be re-phrased if these two are the only relevant studies on the topic. Or at least it should be mentioned whether there are more good quality papers with a similar focus.

lines 267 - 270: this sentence does not make any sense. Are these two separate sentences with omitted punctuation between them?

line 288: did the authors mean "fatigue during COVID-19" ?

lines 303, 305: grammatically incorrect sentences

line 309-317 : grammatically incorrect and incoherent sentences

A dedicated "Limitations" section would be very useful in this paper, as it has several limitations. The authors mention some of these at the end of the "Discussion". The authors should have mentioned the very trivial selection bias, as individuals with symptoms are more likely to volunteer to fill out surveys regarding their condition. The manuscript does not mention if the IT system used for the survey had any measures to prevent repeated submission by the same individual, which would introduce a further confounding factor into the dataset.

Round 2

Reviewer 2 Report

The revised manuscript reads really well. The data is presented in a clear and easily understandable fashion. The limitations are mentioned in adequate detail. I am happy for the manuscript to be published in its current form

Please correct the typo in line 54 : "nervus vagus"